# Improving Well-Being of Farmers Using Ecological Awareness around Protected Areas: Evidence from Qinling Region, China

**DOI:** 10.3390/ijerph18189792

**Published:** 2021-09-17

**Authors:** Li Ma, Yueting Qin, Han Zhang, Jie Zheng, Yilei Hou, Yali Wen

**Affiliations:** School of Economics and Management, Beijing Forestry University, Beijing 100083, China; mali1127@bjfu.edu.cn (L.M.); qinyueting@bjfu.edu.cn (Y.Q.); zhanghan2021@bjfu.edu.cn (H.Z.); zhengjiebjly@163.com (J.Z.)

**Keywords:** ecosystem services, household well-being, perception, China

## Abstract

Sustainability of ecosystems is crucial for improving human well-being and sustainably developing human society. In recent years, global attention towards ecosystems and human well-being has been increasing. Exploring and understanding the relationship between ecosystems and human well-being, and establishing the well-being of residents while protecting the ecosystem have become urgent problems. Based on 618 valid samples collected from communities surrounding seven nature reserves in the Qinling Mountains region of China, this study analyzed the impact of ecosystem services on farmers’ well-being from the perspective of their subjective perception of ecosystem services by using multiple linear regression and seemingly unrelated regression methods. The main conclusions are as follows: supply of vegetation and clean water improves farmers’ well-being, improvement of air quality increases farmers’ life satisfaction, and the sense of belonging and tourism value brought by the ecosystem are important factors for farmers’ physical and mental pleasure and economic benefits. Therefore, the following countermeasures and suggestions are proposed: focusing on establishing the ecological well-being of farmers, improve implementation of the services and benefits provided by the ecosystem to farmers, increase publicity and education to improve the protection consciousness of farmers, and improve community participation mechanisms while mobilizing enthusiasm for protection. This article starts from the perspective of farmers’ perception, attempting to explore whether changes in ecosystem service functions will affect farmers’ well-being, so as to provide new opinions and suggestions for improving farmers’ well-being.

## 1. Introduction

Ecosystem services refer to various service functions and benefits provided by elements of the ecosystem and its ecological processes [1,2,3]. They can be classed into four categories: supply, regulation, cultural, and support services [4,5]. These services are not only key to regional ecological security and sustainable social development, but are also important factors affecting residents’ well-being [6,7]. As ecosystems provide a series of ecosystem services for protected natural areas, their existence has made an important contribution to improving human well-being, promoting sustainable development, and protecting biodiversity [8]. As of the end of 2020, the global coverage of terrestrial nature reserves and reserves has reached 16.64%, and the area of protected land and inland waters has increased from 20.2 million square kilometers in 2010 to the current 22.5 million square kilometers [9]. The ecological environment has improved and biodiversity has become increasingly rich in protected areas through stringent conservation measures [10]. In China, early nature reserves established for the protection of biodiversity did not consider the interests of the indigenous people too much [11]. Due to the restricted use of resources, poverty immediately became synonymous with the communities surrounding the nature reserves. However, in recent years, in order to solve the problem of protection and development of protected areas, the government has introduced a series of protection and development measures to ensure that the well-being of farmers around the protected areas is continuously improved, especially for the poor [12].

Human well-being is a comprehensive, multidimensional, and vague concept that is closely related to people’s living state, perceptions, emotions, etc., and it includes basic material conditions, health, good social relations, safety, freedom of choice and action, and other factors needed to maintain a high quality of life [13,14,15]. Previous studies on well-being include the Life Satisfaction Index [16,17], the Happy Planet Index [18], and the United Nations Human Development Index [19]. There are also some well-being studies, such as the Real Progress Index [20], the Inclusive Wealth Index [21], and the Better Life Index [22]. These consider the impact of nature on human society but are not designed to study ecosystem services and human well-being. A few studies have attempted to incorporate ecological components into the assessment framework when assessing human well-being. For example, the United Nations Millennium Ecosystem Assessment divides human well-being into the following five dimensions: basic living conditions, safety, health, good social relations, and freedom of choice and movement [23]. Summers et al. [24] divided human well-being into basic life needs, economic needs, environmental needs, and subjective well-being. Smith et al. [25] proposed a framework for assessing the well-being of residents in the United States, which is divided into nine dimensions: health, social cohesion, education, safety, living standard, leisure time, spiritual and cultural satisfaction, life satisfaction, and connection with nature. Yang et al. [26], based on the framework of the Millennium Ecosystem Assessment (MA), proposed a set of scale-based human well-being assessment systems, which were used to evaluate the impact of the Wenchuan Earthquake and study the impact of ecosystem service dependence on residents’ well-being [27]. Milner-Gulland et al. [28] proposed a framework for well-being assessment in developing countries; they divided individual well-being into three dimensions: meeting needs, pursuing goals, and quality of life, and suggested that well-being be applied in the performance evaluation of protection policies. Xu Jianying et al. [7] used the Wolong Nature Reserve as an example from the perspective of local residents and studied the relationship among ecosystem services, well-being contributions, and beneficiaries. Hori et al. [29] studied the relationship between ecosystem services and human well-being in coastal areas and compared Canada, China, Japan, South Korea, Russia, and the United States. The MA of ecosystem services and human well-being assessment framework has been widely recognized internationally, and further research has confirmed that the ecosystem and human society interact with each other and have mutually evolved a complex relationship [24,30]. The MA framework explores the social ecological system in the context of understanding and measuring human well-being [31]. These studies show that the understanding of ecology around the world is evolving from natural to human ecology, and increasing attention is being paid to the two-way impact of human well-being and ecosystems. Based on this analysis, MA’s well-being measurement framework can better reflect that well-being comes from ecosystem service functions, hence, this paper chooses this framework for analysis and research.

Ecological perception refers to people’s subjective judgment of whether various services provided by the ecosystem can meet their needs for survival and development. Farmers are the main subjects of economic activities in the ecosystem. Their perception of ecosystem services affects their behaviors while engaging in social and economic activities, which in turn affects the supply of ecosystem services [32]. Studies on the perception of ecosystem services include introduction of research methods [33,34], analysis of stakeholders’ perceptions and preferences [35,36,37], exploration of influencing factors [38,39], and exploration of ecosystem services and residents’ well-being under different socio-economic backgrounds [7,40,41,42,43,44]. Farmers’ well-being is highly subjective and is influenced by both internal and external factors. Under the influence of these factors, farmers’ changes in their own production and lifestyles will affect their evaluation of well-being satisfaction.

The MA report organized by the United Nations points out that human well-being is obviously affected by ecosystem services and clearly indicates a close relationship between ecosystem services and human well-being [45]. Farmers obtain benefits from ecosystem services to meet their needs, which contributes to their well-being. Therefore, ecosystem services serve as an intermediary; they connect ecosystems and human well-being.

In the past, when protection measures for natural protected areas or ecosystems were studied, more attention was paid to the improvement of local ecological benefits. Few people paid attention to the effects of changes in ecological services on people’s well-being while protecting them. Therefore, this article attempts to fill this gap. It is necessary to start from the perspective of farmers’ subjective perception, explore the relationship between the protection effectiveness of protected areas and farmers’ well-being, and explore the key factors that enhance people’s well-being. This paper first analyzed the impact of farmers’ ecosystem service perception on overall well-being, and then further analyzed the impact of farmers’ ecosystem service perception on each well-being index. Based on the feedback of the results, reasonable policy suggestions were put forward to improve the protection effect of protected natural areas and improve farmers’ well-being. The structure diagram of the article is shown in Figure 1.

## 2. Materials and Methods

### 2.1. Study Area

Shaanxi Province is located in the inland northwest of China, between 105°29′–111°15′ E and 31°42′–39°35′ N, covering 10 cities, 107 districts and counties, with a total area of 205,000 square kilometers. The terrain is high in the north and south, and low in the middle. From north to south, it can be divided into three major geomorphological units: the Loess Plateau, the Guanzhong Plain and the Qinba Mountains [46]. The climate varies greatly from north to south. It straddles the three climatic zones of mid-temperate, warm temperate and northern subtropical zone. The climate is continental monsoon, with a multi-year average temperature of 11.6 °C and a multi-year average rainfall of 653 mm. The complex topography and climatic environment have nurtured the rich and diverse habitats of animals and plants in Shaanxi [47,48], and it is one of the important provinces that cherish the distribution of endangered wild animals and plants in the country.

The Qinling region is one of the most typical regions of the ecological system in China. The relationship between ecological protection and regional development is complex and prominent. To effectively protect the habitats of various rare, wild animals and provide them with a good living environment, nature reserves for giant panda protection have been established in the Qinling Mountains of China. Since 1965, when the first Taibai Mountain Nature Reserve was established in Shaanxi Province, 32 various conservation areas have been established in the Qinling Mountains covering a total area of 5591 square kilometers. This accounts for 1/10th of the total area of the Qinling Mountains in Shaanxi Province, forming the large Qinling Mountains Nature Reserve Group. Among them, 16 reserves mainly focus on protecting giant pandas, and about 76 percent of wild giant pandas and 56 percent of giant pandas’ habitats are strictly protected [49]. The number of giant pandas in the Qinling Mountains has increased by 217% from 109 to 345 in the 1980s, the highest in China, according to the country’s fourth giant panda survey. With an average of 10 pandas per 100 square kilometers, the wild population density of giant pandas in the Qinling Mountains ranks first in China. It can be seen that biodiversity conservation in Qinling region has achieved remarkable results [50]. The clear and strict protection policy of the protected area is also one of the reasons for the difference in the perception of ecosystem services between the communities surrounding the protected area and other communities. For example, in terms of ecological restoration, if the environment is damaged, relevant departments will be required to rectify this within a time limit; in terms of village construction, the development and construction of indigenous communities must be coordinated with the local environment, and illegal buildings must be demolished; in terms of industrial and agricultural development, activities such as logging, grazing, hunting, fishing, collecting medicine, reclamation, burning, mining, quarrying, sand digging are prohibited within the scope, but visits and tourism activities can be carried out in the permitted areas of the nature reserve [51]. Additionally, the greatest feature of this area is the large number of people living in protected areas. There are more than 1000 families and nearly 4000 people in the reserve [52]. Therefore, studying the relationship between ecosystems and human well-being in this region is appropriate and is an important reason for choosing this region (Figure 2).

The general nature reserve subjects involved in the study area are shown in Table 1.

### 2.2. Data Collection

The data were obtained from a survey conducted by the research team on rural households around seven protected areas in the Qinling Mountains, China, from August to October 2018. In order to ensure the rationality and typicality of the study area selection, the team members referred to a large amount of second-hand information, including the internet, regional yearbook statistical yearbook, reserve overall planning, etc. on the basis of repeated discussions and consultations with reserve departments. The team carried out preliminary research on the area of research, with a probabilistic risk assessment (PRA) approach to focus group interviews. We randomly selected eight farmers in the village, most of whom we met on the road, and then organized them into a room in the forestry station. Since the farmers were busy with farming, we conducted 6 focus group interviews with 8 people in each group; a total of 48 farmers were interviewed. The interview process was to identify ourselves as the moderators and question the farmers according to the pre-listed outlines on the well-being of farmers and their perception of ecosystem services. Fellow scholars recorded the entire process to facilitate the collation of interview content afterwards. Then, the questionnaire was designed in a targeted manner. The indicators of the questionnaire were designed with reference to the research results of predecessors and combined with the characteristics of the research area. At the same time, this study also invited relevant experts and scholars from the State Forestry and Grassland Administration of China, the Forestry Department of Shaanxi Province, and the investigated nature reserves. These experts are all familiar with ecosystem services or nature reserve management. We invited a total of 12 experts and scholars. The method of conducting expert interviews was to ask the experts to comment on the scale of our research content and direction after introducing our demands, and asking them to introduce the overall situation and characteristics of the Qinling area in Shaanxi, and then collect solutions from the experts for the difficult problems and indicators that were difficult to quantify in the process of collecting data. Finally, the survey questionnaire we designed was given to the experts for review. In this process, the feedback of experts was recorded in detail to correct errors and out-of-time points in the questionnaire design [53].

Regarding early research, the research selected six graduate students with rich experience in social surveys, and three reserves: the Changqing National Nature Reserve, Huangbaiyuan Protected National Nature Reserve, and the Old County National Nature Reserve were chosen as the locations for the preliminary investigation. Then, 180 valid samples were collected in August 2017 to conduct our investigation and research. Based on the pre-survey samples, we conducted several targeted group discussions. Finally, we synthesized all the opinions and suggestions and determined the final questionnaire. Compared with the original questionnaire, the final version of the questionnaire added two location feature indicators, optimized the expression of farmers’ perceptions of ecosystem services, removed issues such as agricultural and forestry production conditions that were not applicable to the research in this article, and removed some well-being variables that would be difficult for farmers to understand. The final questionnaire mainly included farmers’ personal and family basic characteristics, location factors, well-being measurement indicators, and farmers’ subjective perception indicators of ecosystem services (see Appendix A for details). Our research team consisted of 14 experienced graduate students from Beijing Forestry University and staff from the National Natural Reserve Administration. The sampling of sample farmers adopts a combination of group sampling and random sampling. First, according to the level, type, specific natural environment, and location of the protected area, 7 protected areas in the Qinling area were selected as the research area; secondly, according to the level of economic development, the communities in the nature reserve were ranked and divided into equal parts according to the per capita annual income into two groups, then 2 communities were randomly selected from each group; that is, 4 communities were selected in each protected area, and finally about 25 farmers were randomly selected in each community for investigation. After being recommended by the village cadre of the village, interviews were conducted with adult family members who knew the family situation at the homes of the farmers. The interview time was about 1–2 h. The researchers first introduced the purpose of the survey and the questions to the farmers in detail to minimize the potential influence caused by farmers’ incomprehension or misunderstanding when answering questions. Initially, a total of 648 samples were distributed, and invalid samples were eliminated; the final number of valid samples was 618, and the questionnaire effective rate was over 95%.

### 2.3. Measurement of Household Well-Being

#### 2.3.1. Indicator Selection of Household Well-Being

Based on the well-being framework proposed by MA and the view that subjective satisfaction can reflect well-being as proposed by Costanza et al. [1], this study built an indicator system for measuring farmers’ well-being (Table 2) in combination with the characteristics of the Qinling region. A comprehensive index evaluation method was adopted to evaluate well-being using various dimensions. In this study, a well-being evaluation index system was constructed using four dimensions. As some indicators are difficult to measure using actual data but needed to be measured using subjective analysis, a comprehensive evaluation model was introduced to tackle this problem.

By referring to extensive literature, this study comprehensively considered the selection principles of well-being indicators and actual conditions of data acquisition, designed a four-dimensional well-being measurement framework, selected 11 indicators that are frequently used in the current literature, and constructed an evaluation system for farmers’ well-being. The four dimensions were material conditions, health, social relations, and safety. The specific index systems are presented in Table 2. Among them, the subjective index was measured on a scale of 1–5; the higher the number, the higher the satisfaction. After the objective indicators were calculated, the scores were assigned from 1–5 through transformation.

#### 2.3.2. Family Well-Being Index Measurement Method

A comprehensive index evaluation method was proposed to comprehensively evaluate the well-being through each dimension [54].

Its evaluation formula is as follows:(1)$$S=\sum_{j=1}^{n}w_{j}S_{j},$$
where S represents the overall score of well-being, n represents the number of indicators, j represents each indicator of well-being, $w_{j}$ represents the weight of the indicator, and $S_{j}$ represents the normalized value of indicator *j*.

As for the calculation of each weight, the analytic hierarchy process used in previous studies did not exclude the interference of human factors, and the evaluation results were not objective enough. Therefore, the entropy weight method was adopted in this study to calculate the specific weight of each index [55]. The larger the information provided by the index in the system, the smaller the uncertainty, the smaller the entropy, and the higher the weight. Conversely, the weight decreases. Calculation steps are as follows.

First, the standardization of original data: since the indicators designed in this study have different influences on farmers’ well-being, it is necessary to carry out positive processing before standardization [56].

Positive indicators:(2)$$X_{ij}^{\prime }=\frac{X_{ij}-minX_{j}}{maxX_{j}-minX_{j}}\;,$$

Negative indicators:(3)$$X_{ij}^{\prime }=\frac{maxX_{j}-X_{ij}}{maxX_{j}-minX_{j}}\;,$$

Positive indicators include all well-being indicators except “the proportion of medical consumption in total expenditure”, and negative indicators are “the proportion of medical consumption in total expenditure”.

After positive processing, the linear normalization method was adopted to standardize the data.
(4)Yij=Xij′∑i=1mXij′ ,

Second, the index information entropy calculation:(5)$$e_{j}=-k\sum_{i=1}^{m}Y_{ij}lnY_{ij}\;,$$
where the size of *k* depends on the number of samples, $k=\frac{1}{lnm}$; 0 ≤ $e_{j}$ ≤ 1; if $Y_{ij}=0$, then 0.0001 is used instead.

Third, weight calculation:(6)$$w_{j}=\left(1-e_{j}\right)/\sum_{j=1}^{n}\left(1-e_{j}\right),$$

In the formula, $X_{ij}$ represents the value of the *j^th^* evaluation index of the *i^th^* questionnaire, $maxX_{j}$ and $minX_{j}$ represent the minimum and maximum values of the *j^th^* index in all questionnaire data, where *m* represents the number of questionnaires, and *n* represents the number of indicators.

### 2.4. Measurement of Ecosystem Services Awareness

The ecological environment is closely related to the survival and development of human beings, and the acquisition of human well-being is affected by a variety of factors. Therefore, it is necessary to comprehensively consider a variety of circumstances to explore the factors influencing farmers’ well-being. In this study, the effects of ecosystem service perception on farmers’ well-being were analyzed using a seemingly unrelated regression model, and STATA statistical software was used to perform multiple regression analysis.

The dependent variable in this study uses the overall well-being index calculated above. Since the well-being index is a continuous variable and the independent variable is the farmer’s ecosystem service perception variable, and 10 indicators of ecosystem services are selected for supply services, regulation services and cultural services, this study constructed a multiple linear regression model for analysis. The basic formula of the model is as follows [57]:(7)$$Y=\beta_{0}+\beta_{1}X_{1}+\beta_{2}X_{2}\ldots \;\ldots +\beta_{n}X_{n}+\varepsilon_{\alpha },$$
where $\beta_{0}$, $\beta_{1}$ …… $\beta_{n}$ is the undetermined parameter, *n* is the number of explanatory variables, and $\varepsilon_{\alpha }$ is the random error term.

If $b_{0}$, $b_{1}$, $b_{2}$ …… $b_{n}$ are fitted values of $\beta_{0}$, $\beta_{1}$ …… $\beta_{n}$, then the regression estimation equation is
(8)$$\hat{y}=b_{0}+b_{1}x_{1}+b_{2}x_{2}+\ldots b_{n}x_{n}\;,$$
where, $b_{1}$, $b_{2}$ …… $b_{n}$ is a partial regression coefficient, and $b_{0}$ is a constant.

The correlation between the disturbance terms of the equation was considered in the estimation process; thus, the estimation efficiency of the equation was improved. Therefore, to analyze the factors influencing the well-being of farmers obtained from different ecosystem services, this study adopted a seemingly unrelated regression model to measure the impact of ecosystem service perception on farmers’ well-being from different perspectives.

If the four aspects of farmers’ well-being: material conditions, health, social relations, and security are defined as dependent variables, it would allow three variables in the equation to have certain differences; however, for any individual farmer the same unobservable factors may affect all the four aspects of well-being at the same time. Thus, the disturbances of these three equations are likely to be relevant. In contrast to the analysis method of a single regression equation, the quasi-uncorrelated regression model considers that there may be some internal relationships among the explained variables of each equation in the same system. By considering the correlation between the equations, the estimation efficiency can be improved by constructing a system of equations using simultaneous linear regression equations for each group. Therefore, this study used a model to analyze the factors affecting the well-being of farmers. The dependent variable is the farmers’ well-being index while the farmers’ perception of ecosystem services is the main research variable, and the individual characteristic variables and location factors are the main control variables. The equations were constructed as follows:(9)$$Y_{1i}=\beta_{0}+\beta_{i}RL_{1i}+\beta_{2}X_{1i}+\beta_{3}X_{1i}^{\prime }+\varepsilon_{i},$$
(10)$$Y_{1i}=\beta_{0}+\beta_{i}RL_{1i}+\beta_{2}X_{2i}+\beta_{3}X_{2i}^{\prime }+\mu_{i}a,$$
(11)$$Y_{1i}=\beta_{0}+\beta_{i}RL_{1i}+\beta_{2}X_{3i}+\beta_{3}X_{3i}^{\prime }+\gamma_{i},$$

In the above equation, $Y_{1i},\;Y_{2i},\;Y_{3i}$ represent the well-being index of farmer *i*, respectively; $RL_{1i},\;RL_{2i},\;RL_{3i}$ are the vector groups of household ecosystem service perception variables. $X_{1i},\;X_{2i},\;X_{3i}$ are vector groups of personal characteristic variables;$\;X_{1i}{}^{\prime },\;X_{2i}{}^{\prime },\;X_{3i}{}^{\prime }$ are vector groups of location variables, $\beta_{0},\;\beta_{1},\;\beta_{2},\;\beta_{3}$ are coefficient vectors, respectively; $\varepsilon_{i},\;\mu_{i},\;\gamma_{i}$ represent the perturbation term of the equation.

Farmers’ perceptions of the services they receive from the ecosystem are different, and its effects on household well-being variables are inconsistent. Based on existing research, combined with the actual survey and the availability of data, this study selected three possible factors affecting farmers’ well-being: farmers’ perception of ecosystem services (core variable), individual characteristic variables, and location factors (control variables). The descriptive statistics of the variables’ factors are listed in Table 3.

## 3. Results

### 3.1. Comprehensive Evaluation of Farmers’ Well-Being

According to the entropy method, the weight of various indicators of farmers’ well-being were calculated. The calculated results are shown in Table 4, and the weight of material conditions, health, social relations, and safety were 0.16, 0.44, 0.32, and 0.08, respectively. The results show that the contribution rate of each indicator in the criterion layer ranked from large to small is as follows: health, social relations, material demand, and security. The largest proportion of health well-being indicates that health well-being is the most important evaluation index in the criterion layer. It also indicates that the respondents paid significant attention to their own physical health compared to other dimensions of well-being.

Through calculation, the specific scores of each index and target layer in the evaluation criterion layer of farmers’ well-being around the Qinling Giant Panda Reserve were obtained, as shown in Table 5.

The results show that the final score of the farmers’ well-being in this region is 1.83, which indicates that the farmers’ overall well-being is not high; in fact, it is at a lower level than ‘medium’. The scores of the secondary indicators selected in this study, including material conditions, health, social relations, and safety, were 2.69, 1.55, 2.18, and 2.37, respectively. Among them, the evaluated value of the health indicator is in between ‘relatively unsatisfactory’ and ‘general’, while the evaluated value of other indicators is in between ‘moderate’ and ‘relatively satisfactory’. Simultaneously, the scores of the secondary indexes were in the following order: material condition well-being > safety well-being > social well-being > health well-being.

### 3.2. The Impact of Ecosystem Service Perception on the Overall Well-Being of Farmers

It can be seen from the results of the multiple linear regression analysis that F = 6.38, *p* = 0.000 < 1%. Overall, the regression coefficient of the model was significant, indicating that the model fitting effect was good. The regression results of the model are presented in Table 6.

Based on the regression results, this study focused on analyzing the impact of farmers’ perception of ecosystem services on their well-being. Based on the result for the influence of the supply service perception on overall well-being, it can be seen that the ability of the ecological system to provide clean water and vegetation is one of the most important factors affecting farmers’ overall well-being. This indicates that water sources and vegetation are important for establishing farmers’ well-being [58]. The ecosystem helps secure sufficient water sources, vegetation, and other material conditions that improve farmers’ satisfaction with their overall well-being. From the results of the impact of the perception of regulated services on overall well-being, it can be seen that the improvement of air quality has a significant impact on the overall well-being of farmers. The better the air quality, the higher the satisfaction with well-being, which also indicates that the farmers desire increasingly higher environmental standards and green spaces [59]. From the results of the impact of cultural service perception on overall well-being, it can be seen that the sense of belonging and the value of ecotourism are the most significant factors affecting the overall well-being of farmers. It can be seen that the ecological system brings physical and mental satisfaction, and consequently, cultural well-being to members of local rural households [60]. Additionally, the development of tourism boosts the local economy. Additionally, in terms of personal and family characteristics, one’s own physical condition has the most significant influence on well-being, and the better the physical condition, the higher the well-being of a farmer’s family. Second, the distance from the market at the regional level has a general positive impact on the overall well-being of farmers. The further the farmers live from the market, the worse the traffic conditions are, and the information and resources they obtain is limited, so the overall evaluation of well-being satisfaction becomes relatively low.

### 3.3. Impact of Ecosystem Service Perception on Farmers’ Material Well-Being

The results are listed in Table 7. In terms of perception of ecosystem service supply, the respondents’ perception of ecosystem service supply has a negative relationship with their well-being due to material conditions. Families whose main source of income is collecting wild vegetables and herbs are not satisfied with their well-being. This shows that the level of economic development of farmers with high environmental dependence is relatively backwards [61]. In terms of ecosystem regulation service awareness, the higher the farmers’ evaluation of air quality, the lower the satisfaction with housing conditions. This is due to air quality being directly related to the dependence of people’s movement and transportation. Areas with good air quality have relatively sparse transport networks and the convenience of living in such places is restricted. In terms of the perception of ecosystem cultural services, landscape appreciation and the value of ecotourism are important conditions for the development of cultural industries in a region, which have a radiative effect on regional economy and infrastructure. The results show that the higher the evaluation of ornamental landscapes and ecotourism value, the higher their income satisfaction and housing conditions. Simultaneously, a higher rating for the value of ecotourism means there is a higher possibility for carrying out ecotourism, indicating that these families attach more importance to spiritual and cultural satisfaction. Therefore, the proportion of tourism in all consumption types of these families may be higher, while the proportion of food expenditure to meet the basic survival needs may be relatively lower.

### 3.4. Impact of Ecosystem Service Perception on Household Health and Well-Being

The results are listed in Table 8. In terms of the perception of ecosystem supply services, the proportion of medical expenditure of the respondents was significantly reduced by clean water, and the evaluation level of their own physical condition was higher, which indicates the importance of the ecological environment in improving residents’ physical and mental health. Simultaneously, increases in vegetation can significantly improve farmers’ satisfaction with the ecological environment in the village. In terms of the perception of ecosystem regulation services, the improvement of air quality by ecosystem regulation services can significantly reduce the proportion of medical consumption in the total expenditure of farmers and improve their health and the overall ecological environment. Sense of belonging plays an important role in the perception of ecosystem services. In social psychology research, sense of belonging refers to the emotional experience of closeness and pride generated by an individual by the virtue of being classified as a certain unit. A sense of belonging has an important impact on a person’s physical and mental health. This study found that the stronger the sense of belonging among the respondents, the higher their own body condition satisfaction evaluation.

### 3.5. Impact of Ecosystem Service Perception on Farmers’ Social Relationship Well-Being

The results are listed in Table 9. In terms of the perception of ecosystem supply services, the more sufficient the material conditions that farmers enjoy from the ecosystem, the more stable their lives will be, which often results in higher satisfaction with social relations. In terms of the perception of ecosystem regulation services, improvements in air quality, reductions of natural disasters, and the reduction of pests and diseases increase the satisfaction of farmers with their well-being in terms of social relationships. Among them, the reduction of natural disasters, diseases, and insect pests are not only functions of ecosystem regulation, but are also inseparable from the collective governance of the village as a unit. Therefore, farmers have a high satisfaction evaluation of village cadres’ election. In terms of the perception of ecosystem cultural services, the stronger the perception of belonging, esthetic value, and ecotourism value, the higher the satisfaction of farmers’ well-being in terms of social relationships. The cultural services provided by the ecosystem include landscape appreciation, a sense of belonging, and ecotourism. A strong sense of belonging enables farmers to have good neighborhood relationships and deepen their trust with their neighbors. Development of the tourism industry has strengthened cooperation among farmers and plays an important role in neighborhood relations.

### 3.6. Impact of Ecosystem Service Perception on Household Safety Well-Being

The results are listed in Table 10. In terms of the perception of ecosystem services, the more favorable the respondents’ perceptions of the vegetation and water resources provided by the ecosystem, the higher the evaluation index of security and well-being. These energy sources can meet the basic subsistence needs of farmers. In other words, nature does not necessarily provide less monetary value for humans than the economy does [62]. In addition, the landscape created by vegetation and water resources can create conditions for tourism development, so farmers feel that their security well-being are satisfied. In terms of ecosystem regulation service awareness, farmers who perceived the ecological system favorably in terms of air quality improvement and decreases in plant diseases and insect pests had a higher degree of satisfaction with social security. Comparatively, this indicates that regulating ecosystems enables farmers to have suitable conditions for production and living. People’s lives are guaranteed, and thus, social stability is achieved. Simultaneously, improvement of the environment improves the physical quality of farmers, and reduces their dependence on medical interventions to an extent. In terms of the perception of ecosystem cultural services, the respondents’ perceptions of the ecosystem’s aesthetic value and tourism value have a significant positive correlation with satisfaction with health. This is consistent with the phenomenon that people relax their bodies and minds by traveling and appreciating beautiful scenery in recent years. Physical and mental health through natural recuperation in nature reduces the demand for medical treatment to some extent, and then affects the evaluation of farmers on their existing medical conditions.

## 4. Discussion

### 4.1. Analysis of Well-Being Measurement Results

It can be seen from the measurement results of well-being that the score of material condition well-being was the highest, and the score of health and well-being was the lowest. This indicates that the satisfaction of income and material needs are most important for improving farmers’ well-being. Meanwhile, harmonious social relationships and safety well-being were rated relatively highly. In this evaluation system, the score of health well-being was the lowest. In one interview, the interviewee said, “We now live in a better condition, and the poorest people in the village also have food to eat and a place to live, because the state will pay to maintain their basic lives. The neighbors help each other and trust each other. There hasn’t been any theft in the village for a long time.” Through interviews, we know that most farmers believe that the current living conditions and standards can meet their basic survival needs, and the whole village has formed harmonious social relations under the management of the villagers’ self-governance system, which may be the reason for the high scores of material conditions, social relations and safety and well-being. In the interview, some farmers also said, “Some booths in scenic spots are rented to private companies from other places. They throw garbage and discharge sewage while operating, and do not pay attention to protecting our environment.” As you can see from this interview, some farmers in the interview believed that unreasonable resource utilization in the protected areas also led to prominent environmental problems, which threatened the health of farmers. In addition, if someone in a household were to fall ill, the proportion of medical expenses would be high, which places a great economic burden on the household.

### 4.2. Analysis of Regression Results

Judging from the return results of farmers’ well-being, to improve the well-being of farmers, that is, to improve the well-being at various levels, such as material, health, social relations, and safety through the utilization of the ecosystem, better implementation of the services and benefits provided by the ecosystem to farmers is required. Well-being comes from various services provided by the ecosystem, and the individual’s perception of ecosystem services is an intuitive manifestation of the evaluation of well-being, while the perception of the importance of ecosystems and their protection is the basis for participation in protective behavior [7]. The Qinling Nature Reserve is surrounded by a good ecological environment, and the services provided by the ecosystem play an important role in ensuring adequate vegetation, improving infrastructure, and promoting the development of cultural industries, thereby reducing poverty in the region and achieving the “win-win” goal of ecological conservation and improving well-being. Recognition of the complex relationship between the ecological environment and human well-being is conducive to accurately understanding ecological protection and well-being. Therefore, awareness and knowledge of ecological protection concepts should be strengthened so that farmers can correctly understand the relationship between ecological protection and well-being, and realize that participation in ecological protection is closely related to their own economic interests and improvement of well-being.

In the interview, we also asked farmers what they thought or wanted to improve regarding their well-being. They said, “They hope to develop tourism, so that their income will increase”, Some farmers asked, “Can the government raise the compensation standard for Non-commodity Forest?”. Another respondent asked, “Is it possible to arrange some jobs for us in protected areas, such as forest rangers, so that we can have some source of income?”. Combining the regulations on protected areas and the requirements for improving the well-being of farmers, the following measures can be taken to improve the well-being of farmers. Measures to improve the material conditions of farmers include accelerating the development of a green economy, forming a green development mode and lifestyle, building a green industrial structure, and increasing the supply of high-quality green agricultural and forestry products [63]. Additional measures that could be taken includes: raising the level of production technology in rural areas; rational planning, development, and utilization of various ecological resources, such as the development of ecological tourism [64], providing more employment opportunities for ecological conservation, such as jobs in protected areas; improving the ecological compensation system, improving the compensation standard, and ensuring that farmers’ interests are not damaged [65].

### 4.3. Limitations and Future Research

From the perspective of farmers’ subjective perception, this paper proves that farmers’ perception of ecosystem services will affect farmers’ well-being satisfaction, theoretically enriches the contents that affect farmers’ well-being, and in practice explores a new perspective to improve farmers’ well-being. However, it does not mean that the author denies the previous relevant research and its guiding significance for solving practical problems. This article is only an empirical study carried out by the author following in the footsteps of their predecessors. It is a tentative exploration to explain the problem from another angle, and the purpose is to find other factors that improve the well-being of farmers.

In this paper, the selection of farmers’ well-being used mostly qualitative indicators, and was not a complete MA framework. In the future, research should focus on the selection of quantitative indicators to make the indicators more convincing. In addition, this study only involves the data of farmers in protected areas in Shaanxi Province, and the results may not be universally applicable. In the future, the study area should be expanded to make the results more typical and representative, so that they can be replicated in other environments and countries and can be compared.

## 5. Conclusions

With increasing demand for well-being, in a background of high dependence on the ecosystem, understanding factors affecting the well-being of farmers who are both users and protectors of the ecosystem, and the reasonable protection and utilization of the ecosystem have become crucial for establishing well-being [24]. Therefore, in order to determine the differences in peasant household well-being and influencing factors, this study used an unrelated regression model, focusing on the analysis of the direction and degree of impact of ecosystem service perception on farmers’ well-being. Hence, relevant departments can be assisted in understanding the factors and influence degree of farmers’ well-being from the perspective of farmers’ perceptions of ecosystem services and basic characteristics of individuals and regions. Thus, capital ownership and utilization of farmers and the implementation of policies can be adjusted to improve the well-being of farmers.

A multivariate linear model was used to explore the impact of farmers’ subjective perceptions of ecosystem services on their overall well-being. The results show that farmers’ overall well-being is affected by various factors. From the perspective of farmers’ awareness of ecosystem services and vegetation, the supply of clean water improves farmers’ well-being and improvement of air quality increases farmers’ life satisfaction. The sense of belonging and tourism value brought by the ecosystem to farmers are important factors for their physical and mental pleasure and economic benefits. Among the basic characteristics of individuals and families, significantly influencing factors are the physical health of farmers and the distance of their residence from the market.

Using seemingly unrelated regressions to analyze the effects of ecosystem service perception on various elements of well-being, the following conclusions were drawn. First, from the aspect of ecosystem supply service awareness, the more important the supply of ecosystem service function to farmers, the higher their satisfaction with health, social relationships, and security, while material conditions of satisfaction were relatively lower. Particularly, material conditions such as clean water and vegetation provided by ecosystems improved the quality of farmers’ lives and increased their satisfaction with all aspects of well-being. While the farmers with increased collective income had more income, the number of farmers with this income source was relatively small with an unstable income; hence their satisfaction with their material conditions was relatively low. Second, based on the perception of ecosystem regulation services, the higher the air quality evaluation of farmers, the lower their satisfaction with housing conditions. Improving air quality by regulating services significantly reduces the proportion of medical consumption in the total expenditure of farmers, and improves the health of farmers and the overall ecological environment. Farmers with a clear perception of ecosystem regulation services have a relatively high degree of satisfaction with social relationship well-being and can clearly perceive the importance of environmental improvement for survival and social security. Third, from the perspective of the perception of ecosystem cultural services, farmers who pay attention to ecosystem cultural services have different degrees of positive promoting effects on various aspects of well-being compared with those who do not. The higher the evaluation of landscape ornamental value and ecotourism value, the higher the satisfaction with income and housing conditions. The stronger the sense of belonging, the higher the level of satisfaction with their own physical condition. The more obvious the sense of belonging, aesthetic value, and ecotourism value, the higher the satisfaction with social relationship well-being. Additionally, good social and cultural environments reduced the medical demand of farmers and improved their satisfaction with medical conditions.

Well-being not only refers to the improvement of income level or the simple pursuit of the economic value of natural resources, but should also focus on establishing material conditions, health, social relations, and safety obtained by human beings in the ecosystem [41]. Well-being is closely related to the ecosystem, and the realization of well-being can effectively improve its ecological protection behavior [66]. The services provided by good ecosystems around natural protected areas can effectively promote the improvement of local farmers’ well-being [7,42]. However, according to the current perception of farmers regarding ecosystem services, farmers generally do not have a high evaluation of their well-being. Therefore, targeted work should be carried out suited to local conditions. We should not only emphasize what farmers should do and what they are forbidden to do, but let them take initiative in changing their protective behavior while participating in protection and in improving their actual benefits, well-being, and cognition. In conclusion, the improvement of farmers’ well-being and the sustainability of the ecosystem should be integrated into the development planning of the Qinling Protected Area.

## Figures and Tables

**Figure 1 ijerph-18-09792-f001:**
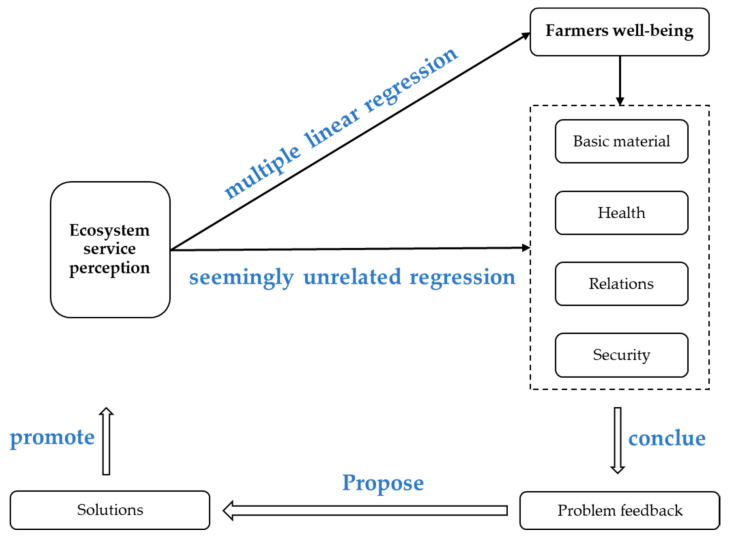
Study the framework diagram.

**Figure 2 ijerph-18-09792-f002:**
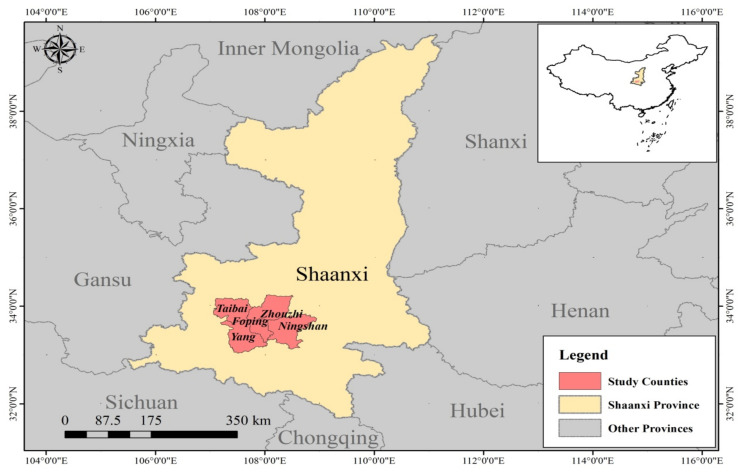
Distribution of the study area.

**Table 1 ijerph-18-09792-t001:** The profile of protected areas in Qinling Mountains.

Name of Reserve	Area (km^2^)	Protection Object	Administrative Region
Foping National Nature Reserve	292.4	Pandas, golden monkeys and other wild animals and their forest ecosystems	Foping county
Zhouzhi National Nature Reserve	563.93	Golden snub-nosed monkey, panda and other wildlife and habitat	Zhouzhi county
Changing National Nature Reserve	299.06	Pandas, takins, musk deer and other wild animals and habitats	Yang county
Huangbaiyuan National Nature Reserve	218.65	Pandas, takin, golden monkey, musk deer	Taibai and Zhouzhi county
Laoxiancheng National Nature Reserve	126.11	Pandas and their habitats	Zhouzhi county
Huangguanshan Provincial Nature Reserve	123.72	Pandas	Ningshan county
Niuweihe Provincial Nature Reserve	134.92	Pandas	Taibai county

The data were obtained from Shaanxi Provincial Reserve data database.

**Table 2 ijerph-18-09792-t002:** The evaluation index system of well-being.

First-Order Index	Second-Order Index (Satisfaction)	Calculation
Basic material	Production and living resources	1 = Very dissatisfied; 2 = Not very satisfied; 3 = neutral; 4 = relatively satisfied; 5 = Very satisfied
	Income	
	Housing conditions	
Health	The proportion of medical consumption in total expenditure	Medical expenditure/total expenditure
	Health status	1 = Very dissatisfied; 2 = Not very satisfied; 3 = neutral; 4 = relatively satisfied; 5 = Very satisfied
	The ecological environment	
Relations	Neighborhood relationships	1 = Very dissatisfied; 2 = Not very satisfied; 3 = neutral; 4 = relatively satisfied; 5 = Very satisfied
	Election Fairness	
	Trust of people around	
Security	Social security	1 = Very dissatisfied; 2 = Not very satisfied; 3 = neutral; 4 = relatively satisfied; 5 = Very satisfied
	Medical conditions	

**Table 3 ijerph-18-09792-t003:** Explanation of variables.

Dimensions of Function	Index Assignment and Meaning	Mean	Standard Deviation
Increased collection revenue	1 = Strongly disagree; 2 = Disagree; 3 = Neutral; 4 = Agree 5 = Strongly agree	2.761	0.914
Increased vegetation cover		2.906	1.145
Clean water was provided		2.856	1.210
Improved air quality		2.847	1.244
Natural disasters have been reduced		2.905	1.020
Reduce pests and diseases		2.871	0.922
Increase the landscape appreciation		2.971	1.039
Generate a sense of belonging		2.934	0.930
Aesthetic value		2.994	0.985
Ecotourism value		2.971	1.038
Respondent gender	1 = male; 0 = female	0.917	0.275
Respondent age	0 = 18 years old and below; 1 = 19–35 years old; 2 = 36–50 years old; 3 = 51–65 years old; 4 = 65 years old and above	3.426	0.990
Highest level of education	1 = primary school and below; 2 = junior high school (technical secondary school); 3 = senior high school (junior college); 4 = bachelor’s degree or above	3.277	1.053
Physical condition	1 = good; 2 = neutral; 3 = mild disease; 4 = major disease	1.506	0.806
Village cadres	1 = yes; 0 = no	0.086	0.280
Main source of revenue	1 = agriculture; 0 = off-farm	0.512	0.500
Living area	1 = protected Area; 2 = outside the protected area	0.298	0.458
Distance to market	1 = 0–10 km; 2 = 10 km–20 km; 3 = 20 km–30 km; 4 > 30 km	2.715	1.487

**Table 4 ijerph-18-09792-t004:** Calculation of weights of well-being indicators.

First-Order Index	Second-Order Index (Satisfaction)	Weight	Equations	Results
Basic material B1	Production and living resources C1	0.564	0.564 × C1 + 0.285 × C2 + 0.151 × C3	0.16
	Income C2	0.285		
	Housing conditions C3	0.151		
Health B2	The proportion of medical consumption in total expenditure C4	0.771	0.771 × C4 + 0.134 × C5 + 0.095 × C6	0.44
	Health status C5	0.134		
	The ecological environment C6	0.095		
Relations B3	Neighborhood relationships C7	0.760	0.760 × C7 + 0.119 × C8 + 0.121 × C9	0.32
	Election Fairness C8	0.119		
	Trust of people around C9	0.121		
Security B4	Social security C10	0.316	0.316 × C10 + 0.684 × C11	0.08
	Medical conditions C11	0.684		

**Table 5 ijerph-18-09792-t005:** Well-being index of famers.

Target Layer	First-Order Index	Index	Second-Order Index (Satisfaction)	Index
Farmers’ well-being 1.83	Basic material	2.69	Production and living resources	2.60
			Income	2.89
			Housing conditions	2.63
	Health	1.55	The proportion of medical consumption	1.34
			Health status	2.43
			Ecological environment	2.02
	Relations	2.18	Neighborhood relationships	2.25
			Election Fairness	2.38
			Trust of people around	1.53
	Security	2.37	Social security	2.02
			Medical conditions	2.53

**Table 6 ijerph-18-09792-t006:** The seemingly unrelated regression estimation results of material well-being.

Explanatory Variables	Coefficient	Standard Error
Increased collection revenue	−0.007	0.008
Increased vegetation cover	0.036 ***	0.011
Clean water was provided	0.024 **	0.012
Improved air quality	0.041 ***	0.012
Natural disasters have been reduced	−0.016	0.011
Reduced pests and diseases	−0.007	0.010
Increased landscape appreciation	0.001	0.012
Generated a sense of belonging	0.023 *	0.012
Aesthetic value	−0.007	0.013
Ecotourism value	0.025 **	0.011
Respondent gender	−0.004	0.024
Respondent age	−0.002	0.007
Highest level of education	−0.002	0.006
Physical condition	−0.018 **	0.009
Village cadres	−0.013	0.023
Main source of revenue	0.003	0.014
Living area	−0.012	0.009
Distance to market	0.012 ***	0.005

* *p* < 0.05, ** *p* < 0.01, *** *p* < 0.001.

**Table 7 ijerph-18-09792-t007:** Seemingly unrelated regression estimation results of material well-being.

Explanatory Variables	Production and Living Resources Satisfaction	Income Satisfaction	Housing Condition Satisfaction
Increased collection revenue	−0.013	−0.023 ***	−0.031 ***
	(0.010)	(0.005)	(0.007)
Increased vegetation cover	−0.006	−0.002	−0.010
	(0.014)	(0.008)	(0.009)
Clean water was provided	−0.006	−0.014	0.002
	(0.020)	(0.011)	(0.013)
Improved air quality	−0.008	−0.003	−0.042 ***
	(0.019)	(0.010)	(0.012)
Natural disasters have been reduced	0.013	0.003	0.015
	(0.017)	(0.009)	(0.011)
Reduced pests and diseases	0.002	0.012	−0.007
	(0.016)	(0.009)	(0.010)
Increased landscape appreciation	0.002	0.016 **	0.026 ***
	(0.012)	(0.006)	(0.008)
Generated a sense of belonging	0.023	0.011	0.019 **
	(0.014)	(0.008)	(0.009)
Aesthetic value	−0.016	0.008	−0.002
	(0.014)	(0.007)	(0.009)
Ecotourism value	−0.018 *	0.030 ***	0.006
	(0.010)	(0.005)	(0.007)

* *p* < 0.05, ** *p* < 0.01, *** *p* < 0.001.

**Table 8 ijerph-18-09792-t008:** Seemingly unrelated regression estimation results of health and well-being.

Explanatory Variables	The Proportion of Medical Consumption	Health Status Satisfaction	Ecological Environment Satisfaction
Increased collection revenue	−0.004	−0.005	0.005
	(0.006)	(0.005)	(0.005)
Increased vegetation cover	0.004	−0.010	0.011 **
	(0.008)	(0.007)	(0.006)
Clean water was provided	−0.012 *	0.018 ***	0.002
	(0.007)	(0.006)	(0.006)
Improved air quality	−0.023 ***	0.013 **	0.012 ***
	(0.006)	(0.005)	(0.005)
Natural disasters have been reduced	0.005	−0.008	−0.006
	(0.006)	(0.005)	(0.005)
Reduced pests and diseases	0.009	−0.009	−0.005
	(0.006)	(0.006)	(0.005)
Increased landscape appreciation	−0.005	−0.002	0.003
	(0.006)	(0.005)	(0.005)
Generated a sense of belonging	0.012	0.024 **	−0.001
	(0.008)	(0.007)	(0.006)
Aesthetic value	−0.007	0.006	0.007
	(0.006)	(0.005)	(0.005)
Ecotourism value	0.011	0.008	0.009
	(0.007)	(0.006)	(0.005)

* *p* < 0.05, ** *p* < 0.01, *** *p* < 0.001.

**Table 9 ijerph-18-09792-t009:** Seemingly unrelated regression estimation results of social relationship well-being.

Explanatory Variables	Neighborhood Satisfaction	Satisfaction with Election Fairness	Satisfaction with the Trust of People Around
Increased collection revenue	−0.004	−0.003	−0.013
	(0.005)	(0.006)	(0.007)
Increased vegetation cover	0.010 **	0.003	0.019 ***
	(0.004)	(0.005)	(0.006)
Clean water was provided	0.010 **	0.015 ***	−0.000
	(0.005)	(0.006)	(0.007)
Improved air quality	−0.000	−0.005	0.022 ***
	(0.005)	(0.006)	(0.007)
Natural disasters have been reduced	0.010 **	0.017 ***	−0.003
	(0.005)	(0.006)	(0.007)
Reduced pests and diseases	−0.008	0.012 **	−0.004
	(0.005)	(0.006)	(0.008)
Increased landscape appreciation	0.001	−0.003	0.007
	(0.005)	(0.006)	(0.008)
Generated a sense of belonging	0.011 **	0.011 *	0.012 *
	(0.005)	(0.006)	(0.007)
Aesthetic value	0.009 *	0.010 *	0.007
	(0.005)	(0.006)	(0.007)
Ecotourism value	0.011 **	−0.002	0.013 *
	(0.005)	(0.006)	(0.007)

* *p* < 0.05, ** *p* < 0.01, *** *p* < 0.001.

**Table 10 ijerph-18-09792-t010:** Seemingly unrelated regression estimation results for safety well-being.

Explanatory Variables	Satisfaction with Public Order	Satisfaction with Medical Conditions
Increased collection revenue	0.003	0.013 **
	(0.005)	(0.005)
Increased vegetation cover	0.012 **	0.010 *
	(0.005)	(0.005)
Clean water was provided	0.019 ***	0.009 *
	(0.005)	(0.005)
Improved air quality	0.009 *	0.013 **
	(0.005)	(0.005)
Natural disasters have been reduced	−0.008	−0.012
	(0.007)	(0.007)
Reduced pests and diseases	0.012 *	−0.003
	(0.007)	(0.007)
Increased landscape appreciation	−0.005	0.007
	(0.006)	(0.006)
Generated a sense of belonging	−0.000	0.008
	(0.006)	(0.006)
Aesthetic value	−0.003	0.012 **
	(0.006)	(0.006)
Ecotourism value	0.003	0.010 *
	(0.005)	(0.005)

* *p* < 0.05, ** *p* < 0.01, *** *p* < 0.001.

## Data Availability

The data presented in this study are available upon request from the corresponding author. The data are not publicly available because of privacy concerns.

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
