# Peer review of "Improving Well-Being of Farmers Using Ecological Awareness around Protected Areas: Evidence from Qinling Region, China"

_ijerph, 2021, doi:10.3390/ijerph18189792_

Round 1

Reviewer 1 Report

This paper examines the impact of farmers' perceptions of ecosystem services on their well-being in the communities surrounding seven nature reserves in the Qinling Mountains region of China. The authors analyze the relationship between ecosystem services and farmers' well-being from multiple perspectives and propose measures to enhance well-being. The study's questionnaire is scientific, the data analysis method is reasonable, and the findings are of practical value. However, there are still some problems, and a major revision is recommended, as follows:、

Abstract

It is recommended to add the significance or value of this study at the end of the ABSTRACT.

Introduction

Line39-41: "……a series of protection measures have continuously improved the welfare of farmers……." This statement is not very accurate. Do all the protection measures have positive effects on farmers' well-being? There may be conservation measures that constrain local development by limiting agricultural or industrial activities. In addition, references are needed to support your points.

Line79-80: The sentence "Hence ......" is more appropriate for the last paragraph of the INTRODUCTION, i. e. the paragraph outlining the objectives and content of the research.

Materials and Methods

Study area: It is suggested that policies and measures for protecting nature reserves and surrounding areas be added to this part. For example, policies on ecological restoration, village construction, industrial development, and agricultural production around the reserves. These policies may be one of the reasons for the differences in ecosystem service perceptions among communities surrounding protected areas and other ones.

Line 168-172: This paragraph is more appropriate in the INTRODUCTION, where you only need to clarify the research method in the METHOD section.

Line 207-208: It is proposed to indicate which indicators are positive and negative in the paragraph constructing the indicator system.

Line221-242: This part would be more appropriate in the RESULTS section.

Line243-250: This part would be more appropriate in the DISCUSSION section.

Line260-263: It is recommended to specify the independent variables in this section instead of vaguely stating " the independent variable contains many different factors".

Results

It is suggested that the RESULTS section contains two parts: farmers' self-rated well-being scores and the impact of ES perceptions on well-being. The results of farmers' well-being in this manuscript are intermixed in the METHOD.

Line303: The data analysis software is more appropriately presented in METHOD.

Line353-365: This paragraph is more appropriately included in the DISCUSSION section as it is not stating the results of the study, but rather some suggestions for well-being enhancement based on the findings, similarly for Line385-398. In addition, this paper lacks a DISCUSSION section, and the authors are advised to add or modify the RESULT section to RESULT AND DISCUSSION.

Author Response

Abstract

1Modification proposal:  It is recommended to add the significance or value of this study at the end of the ABSTRACT.

Response: Thanks for your comment. we have added the significance of this study at the end of the ABSTRACT. "This article starts from the perspective of farmers’ perception, trying to explore whether changes in ecosystem service functions will affect farmers’ well-being, so as to provide new opinions and suggestions for improving farmers’ well-being. "

Introduction

2Modification proposal:

  Line39-41: "……a series of protection measures have continuously improved the welfare of farmers……." This statement is not very accurate. Do all the protection measures have positive effects on farmers' well-being? There may be conservation measures that constrain local development by limiting agricultural or industrial activities. In addition, references are needed to support your points.

Response: We agree with your comments,and we added the research content about the negative impact of protection measures on farmers' well-being, and attached references to the content about the positive impact of protection measures on farmers' well-being. The added content is " In China, the early nature reserves established for the protection of biodiversity did not consider the interests of the indigenous people too much [11]. Due to the restricted use of resources, poverty immediately became synonymous with the communities sur-rounding the nature reserves. However, in recent years, in order to solve the problem of protection and development of protected areas, the government has introduced a series of protection and development measures to ensure that the well-being of farmers around the protected areas is continuously improved, especially for the poor [12].".

3Modification proposal: Line79-80: The sentence "Hence ......" is more appropriate for the last paragraph of the INTRODUCTION, i. e. the paragraph outlining the objectives and content of the research.

Response: The whole paragraph is to illustrate a theoretical analysis of my choice of MA as the framework of the research content, while "Hence ......" is a summary sentence of this paragraph, which explains my choice of MA framework. The reviewer may have misunderstood me because of my unclear sentences, so I have re-concluded the sentences in the article for your better understanding.

Materials and Methods

4Modification proposal: Study area: It is suggested that policies and measures for protecting nature reserves and surrounding areas be added to this part. For example, policies on ecological restoration, village construction, industrial development, and agricultural production around the reserves. These policies may be one of the reasons for the differences in ecosystem service perceptions among communities surrounding protected areas and other ones.

Response: We agree with your comments and add some policy measures for the establishment of the protected area here, including the policy requirements for ecological restoration, village construction, industrial and agricultural and tourism development. The added content is " The clear and strict conservation policy of the protected area is also one of the reasons for the perceived difference of ecosystem services between the surrounding communities and other communities. For example, in terms of ecological restoration, if the environment is damaged, relevant departments will be required to fix it within a time limit; In terms of village construction, the development and construction of indigenous communities should be coordinated with the local environment, and illegal buildings should be forcibly demolished. In terms of industrial and agricultural development, felling, grazing, hunting, fishing, medicinal collection, reclamation, burning, mining, quarrying, sand digging and other activities are prohibited within the strictly protected areas. However, visits and tourism activities may be carried out within the permitted areas of the nature reserves."

5Modification proposal: Line 168-172: This paragraph is more appropriate in the INTRODUCTION, where you only need to clarify the research method in the METHOD section.

Response: We agree with your comment and put it in the INTRODUCTION section.

6Modification proposal: Line 207-208: It is proposed to indicate which indicators are positive and negative in the paragraph constructing the indicator system.

Response: We agree with your comments and have added both positive and negative indicators in the paragraph of the indicator system constructed.

7Modification proposal: Line221-242: This part would be more appropriate in the RESULTS section.

Response: We agree with your comment and put it in the RESULTS.

8Modification proposal: Line243-250: This part would be more appropriate in the DISCUSSION section.

Response: We agree with your comments and add a DISCUSSION section to the structure of the article and put it in the DISCUSSION section.

9Modification proposal: Line260-263: It is recommended to specify the independent variables in this section instead of vaguely stating " the independent variable contains many different factors".

Response: We agree with your comments and point out the selection content of independent variable indicators in this section. The new content is " the independent variable is the farmer's ecosystem service perception variable, and 10 indicators of ecosystem services are selected for supplying services, regulating services and cultural services ".

Results

10Modification proposal: It is suggested that the RESULTS section contains two parts: farmers' self-rated well-being scores and the impact of ES perceptions on well-being. The results of farmers' well-being in this manuscript are intermixed in the METHOD.

Response: Thanks for your advice, we have adjusted the results of farmers' well-being to the RESULTS section.

11Modification proposal: Line303: The data analysis software is more appropriately presented in METHOD.

Response: Thank you for your suggestion, we have adjusted the data analysis software to the METHOD.

12Modification proposal: Line353-365: This paragraph is more appropriately included in the DISCUSSION section as it is not stating the results of the study, but rather some suggestions for well-being enhancement based on the findings, similarly for Line385-398. In addition, this paper lacks a DISCUSSION section, and the authors are advised to add or modify the RESULT section to RESULT AND DISCUSSION.

Response: We agree with your comments and we have added the DISCUSSION part and adjusted these two parts to the DISCUSSION part.

Reviewer 2 Report

This is a very interesting and well-developed paper. However, there are some issues the authors should revise:

- the authors have prepared a proper introduction where they identify previous research and explain how the paper contributes to the current literature. However, they should provide more information in relation to the relevance of the geographical context.

- the topic is very timely and the authors explore the relationship between ecosystems and human well-being from the perspective of farmers in order to protect ecosystem services and residents’ wellbeing.

- in the introduction, the authors should support the following statement with references: “as of May 2021, global protected areas cover nearly 17 percent of the Earth's land area, with more than 22 and 28 million square kilometers of land and ocean designated as natural protected areas or reserves, respectively. The ecological environment has improved and biodiversity has become increasingly rich in protected areas through stringent conservation measures. In China, a series of protection measures have continuously improved the welfare of farmers around the protected areas, especially the low-income population”.

- the description of study area is not supported by previous literature.

- the authors should add sources to the figures and tables.

- in relation to the methodology, the authors should improve the description of the processes in order to allow replicability and validity. The authors should explain the focus group interviews and discussions with experts, how they collected the secondary data, and the questionnaire design. How did the questionnaire change from the initial version?

- the authors should explain why they selected the 20 villages to conduct the interviews.

- in the results, the authors should also add some relevant data from the interviews. They state the interviews lasted between 1 and 2 hours, and direct quotes from the participants would enrich the results and the overall quality of the paper.

- in the conclusion, the authors should identify the limitations of the paper and the opportunities for further research.

Author Response

Reviewer 2:

(1) Modification proposal: The authors have prepared a proper introduction where they identify previous research and explain how the paper contributes to the current literature. However, they should provide more information in relation to the relevance of the geographical context.

Response: Thanks for your suggestion. We have added information about the geographical context in the description of the study area. The new content is " Shaanxi Province is located in the inland northwest of China, between 105°29′-111°15′E and 31°42′-39°35′N, covering 10 cities, 107 districts and counties, with a total area of 205,000 square kilometers. The terrain is high in the north and south, and low in the middle. From north to south, it can be divided into three major geomorpho-logical units: the Loess Plateau, the Guanzhong Plain and the Qinba Mountains [47]. The climate varies greatly from north to south. It straddles the three climatic zones of mid-temperate, warm temperate and northern subtropical zone. The climate is continental monsoon, with a multi-year average temperature of 11.6°C and a multi-year average rainfall of 653 mm. The complex topography and climatic environment have nurtured the rich and diverse habitats of animals and plants in Shaanxi [48,49], and it is one of the important provinces that cherish the distribution of endangered wild animals and plants in the country. "

(2) Modification proposal: In the introduction, the authors should support the following statement with references: “as of May 2021, global protected areas cover nearly 17 percent of the Earth's land area, with more than 22 and 28 million square kilometers of land and ocean designated as natural protected areas or reserves, respectively. The ecological environment has improved and biodiversity has become increasingly rich in protected areas through stringent conservation measures. In China, a series of protection measures have continuously improved the welfare of farmers around the protected areas, especially the low-income population”.

Response: Thanks for your suggestion, we have added references to support our declaration of this part of the content.

(3) Modification proposal: The description of study area is not supported by previous literature.

Response: We agree with your comments and add support from previous literature in the study area section.

(4) Modification proposal: The authors should add sources to the figures and tables.

Response: Thanks for your comments, we have added sources for figures and tables.

(5) Modification proposal: In relation to the methodology, the authors should improve the description of the processes in order to allow replicability and validity. The authors should explain the focus group interviews and discussions with experts, how they collected the secondary data, and the questionnaire design. How did the questionnaire change from the initial version?

Response: Thank you for your advice, we have improved the methods part of the description of the process, this part of the two methods is presented, the first is the measure of farmers' well-being, the second is unrelated regression method, and explains the index meaning of each letter, in addition to the entropy weight method of positive and negative indicators of the corresponding variable name is given out. In the data collection part, focus group interviews and expert discussions are explained in detail, and secondary data collection and questionnaire design methods are supplemented. Compared with the original version of the questionnaire, the final questionnaire optimized the index selection content of ecosystem service perception. The questionnaire questions were designed so that farmers could understand and give correct answers. One well-being Angle in MA framework was deleted, but the index measurement content of farmers' well-being was added to the other four angles.

(6) Modification proposal: The authors should explain why they selected the 20 villages to conduct the interviews.

Response: Thank you for your advice. This part is our writing mistake. We have deleted this part.

(7) Modification proposal: In the results, the authors should also add some relevant data from the interviews. They state the interviews lasted between 1 and 2 hours, and direct quotes from the participants would enrich the results and the overall quality of the paper.

Response: Thank you for your comments. We have extracted key information from the questionnaire interview records and added it into the DISCUSSION section, as follows: ‘Through interviews, we know that most farmers believe that the current living conditions and standards can meet their basic survival needs, and the whole village has formed harmonious social relations under the management of the villagers' self-governance system, which may be the reason for the high scores of material conditions, social relations and safety and well-being. In addition, some farmers in the interview believed that unreasonable resource utilization in the protected areas also led to prominent environmental problems, which threatened the health of farmers. On the other hand, if someone in a household falls ill, the proportion of medical expenses will be high, which brings a great economic burden to the household.’

(8) Modification proposal: In the conclusion, the authors should identify the limitations of the paper and the opportunities for further research.

Response: Thank you for your suggestions. We added a new chapter "Implications, Limitations and Future Research" and discussed the limitations of this paper in this part. The specific content is " From the perspective of farmers' subjective perception, this paper proves that farmers' perception of ecosystem services will affect farmers' well-being satisfaction, theoretically enriches the contents that affect farmers' well-being, and in practice explores a new perspective to improve farmers' well-being. However, it does not mean that the author denies the previous relevant research and its guiding significance for solving practical problems. This article is only an empirical study carried out by the author following the footsteps of the predecessors. It is a tentative exploration to ex-plain the problem from another angle and thinking, and the purpose is to find other factors that improve the well-being of farmers. 

In this paper, the selection of farmers' well-being is mostly qualitative indicators, and it is not a complete MA framework. In the next step, the research should focus on the selection of quantitative indicators to make the indicators more convincing. In addition, this study only involves the data of farmers in protected areas in Shaanxi Province, and the results may not be universally applicable. In the next step, the study area should be expanded to make the results more typical and representative, so that they can be replicated in other environments and countries and can be compared."

Reviewer 3 Report

Dear Authors

It is a great pleasure reviewing your manuscript "Improving well-being of farmers using ecological awareness around protected areas– Evidence from Qinling region, China". Please find attached my feedback for your consideration.

Thank you and kind regards.

Author Response

Reviewer 3:

Introduction

(1) Modification proposal: It is still unclear about the gap/s in the literature that this study aims to fill. This needs to be clearly articulated so that your proposed study is supported.

Response: Thank you for your comments. In the last paragraph of the INTRODUCTION, we describe the gaps in the existing literature that we want to fill in order to support our research. The specific content is " In the past, when protection measures for natural protected areas or ecosystems was studied, more attention was paid to the improvement of local ecological benefits. Few people paid attention to the effects of changes in ecological services on people's well-being while protecting them. Therefore, this article attempts to fill this gap. It is necessary to start from the perspective of farmers’ subjective perception, explore the relationship between the protection effectiveness of protected areas and farmers’ well-being, and explore the key factors that enhance people’s well-being. "

(2) Modification proposal: Consider the provision of a diagram to outline the framework used for this study so that it is clear to readers on what and how the authors are proposing to achieve the aim of this specific study.

Response: Thanks for your comments, we have provided a chart outlining the framework used in this study so that readers have a clear idea of what we propose and how the purpose of this specific study is achieved.

(3) Modification proposal: Last paragraph, fourth sentence on page 2 – “Farmers' perceptions of ecosystem services are important measures of regional changes in ecological services.” Please provide some brief discussion and justifications on why this is the case, using some references to support.  

Response: Thank you for your comment. This sentence is a mistake in our translation, and the translated meaning is somewhat different from what we intended to express. We are trying to improve our English level, and we have revised this part and clearly expressed what we want to study in the last paragraph.

Materials and Methods

(4) Modification proposal: This study appears to have been conducted through a two-stage process; that is Stage 1 – preliminary research conducted with 180 samples in August 2017; and Stage 2 – household interviews conducted with 618 samples between August to October 2018. Perhaps consider having appropriate sub-headings to clearly “sign- post” them so that readers can follow more easily.    

Response: Thank you for your comment. We marked our survey data in the subtitle. As the samples obtained in the preliminary study of the first stage could not fully support this study, we did not use this part of data.

(5) Modification proposal: Section 2.1, detailed information about the study area has been provided but they need to be supported by ap-propriate references. This lack of references is evident in all the sentences prior to the reference #42 made at the back of the paragraph.  

Response: Thank you for your comment. Several related references have been added as required, and corresponding additions have been made in the reference part.

(6) Modification proposal: Section 2.2, first paragraph on page 4 – “The team carried on preliminary research to area of research, the probabilistic risk assessment (PRA) approach to focus group interviews, and then targeted the design of the questionnaire ................. the Shaanxi Provincial Forestry Department, the National Bureau of Forestry and Grassland Nature Reserve.” Please specific how many focus group interviews conducted, and how many experts have been invited for discussion. How are they selected?    

Response: We agree with you. We have described in detail in this section, including how many focus group interviews we conducted, how many experts we invited, and why we selected these experts. In addition to the initial interview, we also introduced in detail the process of our focus group interview and the way to carry out the expert interview.

(7) Modification proposal: Section 2.2, first paragraph, second last sentence on page 4 – “Finally, the questionnaire was determined based on all opinions and suggestions.” Please clarify what have been modified to the final questionnaire used for the household interviews?     

Response: We agree with your opinion. In this part, we introduce the modifications made in the final version of the questionnaire in detail. For example, compared with the original version, two location characteristic indexes are added to the final version of the questionnaire, the way of expression of farmers' ecosystem service perception is optimized, and questions that are not applicable to this study such as agricultural and forestry production are deleted. Some variables that would cause ambiguity and be difficult to understand for farmers in well-being measurement were deleted.

(8) Modification proposal: Section 2.2, second paragraph, second last sentence on page 4 – “We adopted the method of random encounter sampling and conducted a formal survey through semi-structured interviews and questionnaires.” Please elaborate on how the random sampling is being used to select the villages and households.    

Response: We agree with you that we have made a detailed explanation and supplement on how to select villages and households by random sampling. The specific content is " The sampling of sample farmers adopts a combination of group sampling and random sampling. First, according to the level, type, specific natural environment, and location of the protected area, 7 protected areas in the Qinling area are selected as the research area; secondly, according to the level of economic development, the communities in the nature reserve are ranked and divided into equal parts according to the per capita annual income Two groups, and then randomly select 2 communities from each group, that is, select 4 communities in each protected area, and finally randomly select about 25 farmers in each community for investigation. "

(9) Modification proposal: Section 2.2, second paragraph, last sentence on pages 4 to 5 – “After introduction by the local village head, one-to-one interviews were conducted with the heads of the families, and the interviews were supplemented by other 2 adult members of the family.” This sentence is a little confusing if it is a one-to-one interview or multiple members involved. One-to-one interview versus supplemented by other adult members of the family? In addition, why is the heads of the families interviewed? What is the basis? Please clarify.   

Response: We agree with your suggestion. We have amended this sentence to clarify any misunderstandings. The specific content is "After being recommended by the village cadre of the village, interviews were conducted with adult family members who knew the family situation at the homes of the farmers. The interview time was about 1-2 hours. "

(10) Modification proposal: Section 2.2, first paragraph, last sentence on page 5 – “A total of 618 valid samples were collected, with an effective rate of over 95%.” Please specify how many were initially conducted.

Response: We agree with your suggestion, and we have supplemented the data of the initial sample size in the article. The specific content is " Initially, a total of 648 samples were distributed, and invalid samples were eliminated, the final number of valid samples was 618, and the questionnaire effective rate was over 95%."

(11) Modification proposal: Section 2.3, first paragraph on page 5 – “The MA report organized by the United Nations points out that human well-being is obviously affected by ecosystem services ............... they connect ecosystems and human well-being.” Consider moving the content in this paragraph to Section 1, last paragraph on page 2 as this provide a supporting argument to the importance of ecosystem and farmers’ well- being.

Response: We agree with your suggestion and we have moved this part to the correct position.

(12) Modification proposal: Section 2.3.1, second paragraph, second last sentence on page 5 – “The specific index systems are presented in Table 2. Among them, the subjective index was measured on a Likert scale of 1–5; the higher the number, the higher the satisfaction.” Please consider including the suggestions in red. 

Response: Thank you for your comment. This part of our content is to explain the meaning of each number, which is an explanation of the content of the table.

(13) Modification proposal: Sections 2.3.2 to 2.4 on pages 6 to 8 – the formulas presented are adequately explained but they lack supporting references. Please include appropriate references.    

Response: We agree with your suggestion and have attached appropriate references.

(14) Modification proposal: The “Basic personal and family characteristics” in Table 5 on page 9 is somewhat different to the questionnaire in Appendix A – “The householder age” - less than or equal to 18 is missing; “The highest level of education” is different to those identified in Appendix A. Please correct.

Response: Thank you for your comments and we have corrected the inconsistencies.

(15) Modification proposal: Is there an ethics application for this research to be conducted?  

Response: In our study, participants were invited to join in the survey voluntarily and anonymously without offending their privacy and generating ethic issues. Therefore, we did not seek approval for this case. All human studies where non-routine procedures are not used in this study. Before all interviews, the content of the study was explained to the interviewees and their agreement was obtained.

Implications, Limitations and Future Research

(16) Modification proposal: Consider adding a section heading “Implications, Limitations and Future Research” or similar 

Response: Thanks for your comments, we have added a section heading to the content.

(17) Modification proposal: This section should outline the theoretical and practical contributions based on the findings. This can link back to what and how your identified gap/s (identified at the beginning of this paper) are filled through this study.    

Response: Thanks for your comments, we outline the theoretical and practical contributions of this paper in a new section. The specific content is " From the perspective of farmers' subjective perception, this paper proves that farmers' perception of ecosystem services will affect farmers' well-being satisfaction, theoretically enriches the contents that affect farmers' well-being, and in practice explores a new perspective to improve farmers' well-being. However, it does not mean that the author denies the previous relevant research and its guiding significance for solving practical problems. This article is only an empirical study carried out by the author following the footsteps of the predecessors. It is a tentative exploration to ex-plain the problem from another angle and thinking, and the purpose is to find other factors that improve the well-being of farmers." 

(18) Modification proposal: The limitations (e.g. geographic region, farmers) of the study as well as future research (e.g. replicate in other contexts and countries to make comparison) have not been addressed too.      

Response: Thank you for your comments. We have added the limitations of the article and the prospect of future research in the content. The specific content is "In this paper, the selection of farmers' well-being is mostly qualitative indicators, and it is not a complete MA framework. In the next step, the research should focus on the selection of quantitative indicators to make the indicators more convincing. In addition, this study only involves the data of farmers in protected areas in Shaanxi Province, and the results may not be universally applicable. In the next step, the study area should be expanded to make the results more typical and representative, so that they can be replicated in other environments and countries and can be compared. "

Round 2

Reviewer 1 Report

The manuscript is more organized and easy to read after a round of revision. However, there are still some minor problems in the present manuscript. 
1. The title needs to be more concise;
2. It is recommended that subheadings be added to the “Discussion” section so that readers can more easily understand the main points of the discussion;
3. The “Implications, Limitations and Future Research” is not sufficient as a separate section, and this section can be included in the “Discussion”.

Author Response

Reviewer 1:

1Modification proposal:  The title needs to be more concise;

Response: Thanks for your comment. We finally kept the title unchanged by referring to your opinion and the second reviewer's opinion on the title of the article. “Improving well-being of farmers using ecological awareness around protected areas–Evidence from Qinling region, China”

2Modification proposal:  It is recommended that subheadings be added to the “Discussion” section so that readers can more easily understand the main points of the discussion;

Response: Thanks for your comments, we have added subheadings to the DISCUSSION section.

3Modification proposal: The “Implications, Limitations and Future Research” is not sufficient as a separate section, and this section can be included in the “Discussion”.

Response: Thanks for your comment. We have adjusted the content of "Implications, Limitations and Future Research" to the DISCUSSION part.

Reviewer 2 Report

While the authors have revised the paper, some revisions are:

- the title should be kept as it was because the number of surveys is described in the abstract.

- direct quotes from the participants would improve the overall quality of the paper.

- the authors should expand section 6.

Author Response

Reviewer 2:

1Modification proposal:  the title should be kept as it was because the number of surveys is described in the abstract.

Response: Thanks for your comment, we have changed the title to the original.

2Modification proposal: direct quotes from the participants would improve the overall quality of the paper.

Response: Thank you for your comments, we have quoted the participants and included them in the DISCUSSION.

In one interview, the interviewee said, “We now live in a better condition, and the poorest people in the village also have food to eat and a place to live, because the state will pay to maintain their basic lives. The neighbours help each other and trust each other. There hasn't been any theft in the village for a long time.”

In the interview, some farmers also said, "Some booths in scenic spots are rented to private companies from other places. They throw garbage and discharge sewage while operating, and do not pay attention to protecting our environment."

 In the interview, we also asked farmers what they thought or wanted to improve their well-being. They said, "They hope to develop tourism, so that their income will increase." Some farmers asked, "Can the government raise the compensation standard for Non-commodity Forest?" Another respondent asked, "Is it possible to arrange some jobs for us in protected areas, such as forest rangers, so that we can have some source of income?"

3Modification proposal: the authors should expand section 6.

Response: Thank you for your comment. Considering the brevity of the sixth part and the suggestions of reviewer 1, we have adjusted the sixth part into the discussion.

Reviewer 3 Report

Dear Authors

Thank you for responding to my feedback. Please find below 2 main points for your clarification.

  1. While Figure 1. Study the framework diagram has been added, but there is no reference made to this newly added figure in any part of the manuscript.
  2. The response (lines 174-177) to "(6) Modification proposal" indicated  that 8 farmers in the village were randomly selected whom were met on the road. Is this based on a convenience sampling, select anyone passed by or are there selection criteria used? The response also stated that 6 focus groups were conducted, is this with only 8 randomly selected farmers? This does not appear to be logical for a need to conduct 6 focus groups. Some explanations will be appreciated.

Thank you and kind regards. 

Author Response

Reviewer 3:

1Modification proposal:    While Figure 1. Study the framework diagram has been added, but there is no reference made to this newly added figure in any part of the manuscript.

Response: Thank you for your comment. We have added the content of the frame diagram into the manuscript, the specific content is as follows: "This paper first analyzed the impact of farmers' ecosystem service perception on the overall well-being, and then further analyzed the impact of farmers' ecosystem service perception on each well-being index. Based on the feedback of the results, reasonable policy suggestions were put forward to improve the protection effect of protected natural areas and improve farmers' well-being. The structure diagram of the article is shown in Figure 1. "

2Modification proposal:     The response (lines 174-177) to "(6) Modification proposal" indicated  that 8 farmers in the village were randomly selected whom were met on the road. Is this based on a convenience sampling, select anyone passed by or are there selection criteria used? The response also stated that 6 focus groups were conducted, is this with only 8 randomly selected farmers? This does not appear to be logical for a need to conduct 6 focus groups. Some explanations will be appreciated.

Response: Thank you for your comment. The 8 villagers we met on the road were selected based on convenient sampling. For each villager we met, we would clearly explain the purpose of our research and conduct focus group interviews with them after obtaining their consent. We conducted a total of 6 groups of interviews with 8 people in each group, and a total of 48 farmers were selected. In order to reduce misunderstanding, this part of the text has been revised.
